# Learning Visuotactile Estimation and Control for Non-prehensile Manipulation under Occlusions

**Juan Del Aguila Ferrandis**[1], **João Moura**[1,2], **Sethu Vijayakumar**[1,2]
[1]School of Informatics, The University of Edinburgh, UK
[2]The Alan Turing Institute, UK

**Abstract:** Manipulation without grasping, known as non-prehensile manipulation, is essential for dexterous robots in contact-rich environments, but presents many challenges relating with underactuation, hybrid-dynamics, and frictional uncertainty. Additionally, object occlusions in a scenario of contact uncertainty and where the motion of the object evolves independently from the robot becomes a critical problem, which previous literature fails to address. We present a method for learning visuotactile state estimators and uncertainty-aware control policies for non-prehensile manipulation under occlusions, by leveraging diverse interaction data from privileged policies trained in simulation. We formulate the estimator within a Bayesian deep learning framework, to model its uncertainty, and then train uncertainty-aware control policies by incorporating the pre-learned estimator into the reinforcement learning (RL) loop, both of which lead to significantly improved estimator and policy performance. Therefore, unlike prior non-prehensile research that relies on complex external perception set-ups, our method successfully handles occlusions after sim-to-real transfer to robotic hardware with a simple onboard camera. See our video: `https://youtu.be/hW-C8i_HWgs`.

**Keywords:** State Estimation, Reinforcement Learning, Tactile Sensing, Non-prehensile Manipulation

## 1 Introduction

Non-prehensile manipulation is a crucial skill for enabling versatile robots to interact with ungraspable objects, using actions such as pushing, rolling, or tossing. However, achieving dexterous non-prehensile manipulation in robots poses significant challenges. During contact interactions, different contact modes arise such as sticking, sliding, and separation, and transitions between these contact modes lead to hybrid dynamics [1, 2, 3]. Furthermore, due to its underactuated nature, it requires long-term reasoning about contact interactions as well as reactive control to recover from mistakes and disturbances [1, 2]. The frictional interactions between the robot, the object, and the environment are difficult to model, which creates uncertainty in the behavior of the object [4, 5].

The highly uncertain nature of the underactuated frictional interactions [4, 5] make the non-prehensile manipulation problem especially sensitive to occlusions. Previous non-prehensile works assume near-perfect visual perception from external systems, providing either point-cloud [6] or pose observations [7, 8, 9, 10, 11]. However, moving towards more versatile onboard perception will make frequent occlusions unavoidable, either due to obstacles in the environment, self occlusions, or even human-induced occlusions, for instance in a human-robot collaboration setting.

In this paper, we propose a learning-based system for non-prehensile manipulation that leverages tactile sensing to overcome occlusions in the visual perception. We decompose the learning task into a state estimation and a control task. We begin by training a privileged policy with reinforcement learning (RL) in an occlusion-free environment and periodically storing checkpoints, i.e. the policy's parameters, at various stages of training. Then, by rolling out a wide range of optimal and suboptimal policy checkpoints, we are able to collect a diverse dataset of trajectories, which we pro-

8th Conference on Robot Learning (CoRL 2024), Munich, Germany.

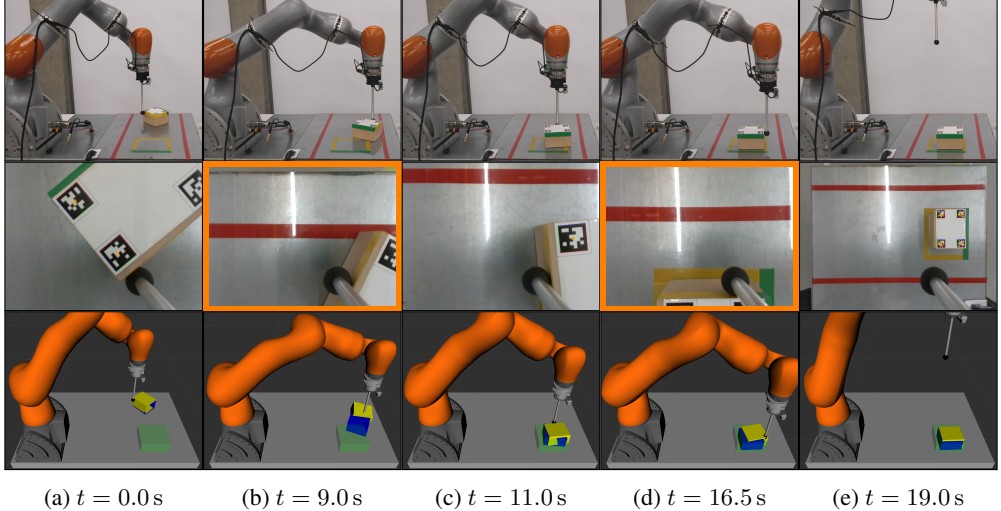

| (a) $t = 0.0$ s | (b) $t = 9.0$ s | (c) $t = 11.0$ s | (d) $t = 16.5$ s | (e) $t = 19.0$ s |

Figure 1: Snapshots from an exemplar robot motion pushing a box to the target with occlusions. Columns correspond to different time-frames, with (b) and (d) corresponding to two snapshots where the robot view occludes the reading of the box pose. The first row shows the robot setup, the second row shows the robot camera view for the tag/object detection, and the final row shows the rviz view with yellow and blue corresponding to the observation (from the camera) and the estimation of the box poses. The green area marks the target.

cess to introduce synthetic occlusions in the visual perception as well as sensory noise. We use this trajectory dataset to train a state estimator within a Bayesian deep learning framework to capture the uncertainty in the state prediction. We note that capturing the uncertainty within such 'black-box' deep learning based state estimators is important for developing more robust downstream policies and enabling safer and more reliable real-world deployment. Finally, we train an RL control policy in an environment with visual occlusions, using the pre-learned state estimator in the loop.

We evaluate our proposed system on a challenging planar pushing task with frequent and prolonged visual occlusions, achieving 94% success rate. We also deploy the learned estimation and control framework on a robotic hardware with zero-shot sim-to-real transfer. Our contributions are:

- We propose a novel approach to learn state estimators based on force interaction for non-prehensile manipulation under visual occlusions, leveraging privileged policies to generate diverse interaction data and a Bayesian deep learning framework to model the uncertainty.
- We also propose to learn uncertainty-aware control policies by integrating the pre-trained state estimator in the RL training loop.
- We show that, by modeling the uncertainty, we are able to learn significantly more accurate estimators and, furthermore, training uncertainty-aware control policies with the estimator in the loop leads to improved performance.
- We validate that our approach exhibits good transferability to a physical robot using purely on-board perception with both naturally-occurring and human-induced visual occlusions.

## 2 Related Work

**Non-prehensile Manipulation.** Previous works successfully learn a wide variety of non-prehensile manipulation skills, including planar pushing [7, 8, 9, 10, 11], pivoting against the external environment to realize grasps [12, 13], retrieving objects in cluttered environments [14, 15], and 6D rearrangement [6, 16, 17]. However, all of these methods avoid visual occlusions, generally by relying on complex perception schemes, external to the robot, that guarantee continuous tracking of the manipulated object. Some works rely on motion capture set-ups, such as Vicon, with multiple specialized cameras such that if some cameras experience occlusions, others continue providing reliable readings [7, 8, 11]. Other works employ vision-based set-ups, with tracking tools such as

AprilTag [18]; however, they rely on external cameras strategically placed to avoid occlusions [9]. Works without explicit tracking systems also require specialized set-ups to avoid occlusions, for instance by using a camera under a transparent table to observe the object [10] or combining multiple external depth cameras to create a point cloud [6, 17]. In contrast, our proposed approach enables us to perform non-prehensile manipulation tasks with a simple on-board camera and leverage tactile sensing to keep solving the task during prolonged visual occlusions. We believe this is an important step towards deploying robotic systems outside of the lab environment, without access to external perception set-ups and, hence, with likely occurrence of occlusions.

**Dealing with Observation Uncertainty.** There are model-based approaches for integrating uncertainty into planning, requiring either analytical or learned dynamics models [19, 20]. More relevant to this work, there are also multiple learning-based approaches for dealing with observation uncertainty. Many works train control policies directly in the simulated target environment and use domain randomization as well as synthetic observation noise during learning to achieve zero-shot sim-to-real transfer [7, 11, 21, 22]. Others combine multiple sensory modalities such as vision, haptic, and audio signals to handle scenarios where one or more modality might be unreliable [23, 24, 25, 26]. Recently, a popular technique is to first learn a teacher policy assuming access to the ground-truth environment state and then train a student policy to imitate the teacher's actions given the observations from the deployed environment [27, 28, 29]. While taking inspiration from these methods, we find that these alone are insufficient to achieve good performance in the challenging non-prehensile manipulation task with prolonged visual occlusions.

## 3 Method

### 3.1 Problem Formulation

We formulate the non-prehensile task as a finite-horizon Partially Observable Markov Decision Process (POMDP) described by the tuple $(\mathcal{S}, \mathcal{A}, \Omega, \mathcal{O}, \mathcal{P}, \mathcal{R}, H, \gamma)$. In particular, at every timestep $t$, the environment has a state $\boldsymbol{s}_t \in \mathcal{S}$ and the agent takes an action $\boldsymbol{a}_t \in \mathcal{A}$, resulting in a new state $\boldsymbol{s}_{t+1}$ according to the transition model $\mathcal{P} : \mathcal{S} \times \mathcal{A} \to \Pr(\mathcal{S})$. Due to the partially observable setting, the agent is unable to perceive $\boldsymbol{s}_{t+1}$ directly and instead receives an observation $\boldsymbol{o}_{t+1} \in \Omega$ that depends on $\boldsymbol{s}_{t+1}$ and $\boldsymbol{a}_t$ based on the observation model $\mathcal{O} : \mathcal{S} \times \mathcal{A} \to \Pr(\Omega)$. The agent receives a reward $r_t$ associated with the transition $(\boldsymbol{s}_t, \boldsymbol{a}_t, \boldsymbol{s}_{t+1})$, which is given by the reward function $\mathcal{R} : \mathcal{S} \times \mathcal{A} \times \mathcal{S} \to \mathbb{R}$. The objective is to learn a policy $\pi : \Omega \to \Pr(\mathcal{A})$ that maximizes the discounted cumulative reward $\mathbb{E}_\pi \left[ \sum_{t=0}^{H-1} \gamma^t r_t \right]$, with maximum horizon $H$ and discount factor $\gamma \in (0, 1)$.

The partially observable nature of the task is critical since we are interested in learning non-prehensile manipulation policies that handle occlusions in the visual perception. Specifically, we assume that, in addition to correlated and uncorrelated Gaussian sensory noise, the environment observation $\boldsymbol{o}_t$ features occlusions in the pose of the manipulated object. We assume that robot end-effector pose and force measurements are available through proprioception without any occlusions.

We decompose the learning task into a state estimation and a control task, where we first learn a state estimator $f(\boldsymbol{o}_t) = (\hat{\boldsymbol{s}}_t, \hat{\boldsymbol{\Sigma}}_t)$ and then use it to learn a control policy $\pi(\hat{\boldsymbol{s}}_t, \hat{\boldsymbol{\Sigma}}_t)$. In practice, since we consider occlusions in the object pose $\boldsymbol{q}_t^{obj}$, we only train $f$ to predict $\boldsymbol{q}_t^{obj}$. Additionally, we assume sequential access to states and observations, enabling the use of recurrent architectures for $f$ and $\pi$ with internal representations that capture the input evolution.

### 3.2 Learning the State Estimator

**Data Collection.** Training a model-free state estimator for non-prehensile manipulation that is robust to visual occlusions demands a large and diverse dataset of interactions to ensure high accuracy and reduce out-of-distribution scenarios. Collecting this data in a real-world setting is difficult due to the extensive data requirements, the cost of hardware, and the need for human supervision – consequently, we utilize simulation instead. This has the added advantage of being able to ex-

plore simulation environments with different physical characteristics, which is crucial to enable the learned state estimator to generalize effectively across different environments. We propose that an effective strategy is to first train a privileged policy $\pi_{priv}(s_t)$ that has access to the ground truth state of the environment, and then rollout a set of $n$ checkpoints stored during training $\{\pi_{priv}^1, \ldots, \pi_{priv}^n\}$ to collect the interaction dataset $\mathcal{D} = \{\Gamma_1, \Gamma_2, \ldots\}$, where $\Gamma_i = \{s_0, s_1, \ldots\}$ denotes a trajectory in the state space generated by one of the privileged policy checkpoints. The privileged checkpoints include untrained random policies as well as a wide range of suboptimal and optimal policies. This approach allows us to collect an arbitrary amount of task-relevant data, ensuring that a diverse set of behaviors is present, and thereby preventing the estimator from becoming biased towards a particular behavior. Note that it is straightforward to use RL for learning $\pi_{priv}(s_t)$ without occlusions.

**Data Processing.** We want to train a state estimator $f$ that maps environment observations $o_t$ to estimated object poses $\hat{q}_t^{obj}$, along with the associated uncertainty $\hat{\Sigma}_t$. Therefore, we process the dataset $\mathcal{D}$ using our assumed observation model $\mathcal{O}$ to produce a dataset $\mathcal{D}' = \{\Gamma_1', \Gamma_2', \ldots\}$, where $\Gamma_i' = \{(o_0, q_0^{obj}), (o_1, q_1^{obj}), \ldots\}$ is a trajectory in the observation space with the corresponding ground-truth trajectory of the object pose. We consider a state $s_t = (q_t^{obj}, q_t^e, f_t^e)$ composed of the object $q_t^{obj}$ and the robot end-effector $q_t^e$ poses, and the end-effector force $f_t^e$. We use an observation model $o_t = \mathcal{O}(s_t)$ that adds both correlated (sampled at the beginning of the trajectory), and uncorrelated (sampled at each timestep) Gaussian noise to the observation of the state. Additionally, $\mathcal{O}$ applies occlusions to the object pose, which can start at every timestep $t$ with a probability of $p$ and have duration sampled from a Gaussian distribution. Throughout an occlusion, the observed object pose remains the same as the last available pose prior to the start of the occlusion. The observation also includes a binary indicator $\xi$, which is $1$ for occlusion and $0$ otherwise.

**Types of Uncertainty in Bayesian Deep Learning.** Our state estimator $f$ captures the two main types of uncertainty modelled under a Bayesian framework: *aleatoric* uncertainty and *epistemic* uncertainty [30]. *Aleatoric* uncertainty captures the noise inherent to the data. This arises for example due to sensory noise and it is irreducible even with additional data. *Epistemic* uncertainty captures the uncertainty in the model parameters, and it can be arbitrarily reduced with sufficient training data. Furthermore, we consider a setting where the uncertainty varies depending on the input to the model, known as *heteroscedastic* uncertainty.

**Modeling Aleatoric Uncertainty.** We use the dataset $\mathcal{D}'$ to learn a state estimator $f(o_t) = (\hat{q}_t^{obj}, \hat{\Sigma}_t^{ale})$ that predicts both object pose and aleatoric uncertainty. We formulate our estimator to predict the mean and covariance, such that $p(q_t^{obj} \mid o_t) \sim \mathcal{N}(\hat{q}_t^{obj}, \hat{\Sigma}_t^{ale})$, and use the negative log-likelihood [30, 31] as the loss function

$$\mathcal{L} = \frac{1}{2} \ln \left( \left| \hat{\Sigma}_t^{ale} \right| \right) + \frac{1}{2} \left( q_t^{obj} - \hat{q}_t^{obj} \right)^T \left( \hat{\Sigma}_t^{ale} \right)^{-1} \left( q_t^{obj} - \hat{q}_t^{obj} \right). \tag{1}$$

In practice, we use a diagonal covariance matrix to reduce the output dimensionality.

**Modeling Epistemic Uncertainty.** To model the epistemic uncertainty of the learned state estimator we use Dropout variational inference [30, 31, 32]. More specifically, dropout [33], a common regularization technique for neural networks, can be interpreted as approximate variational inference in Bayesian neural networks [30, 32]. Hence, we apply dropout to the state estimator $f$ at both training and testing time, which allows us to perform stochastic forward passes, effectively sampling from the approximate posterior of $f$. This is known as Monte Carlo (MC) dropout. Suppose that we perform $M$ stochastic forward passes of $f$, we can approximate the epistemic uncertainty as

$$\hat{\Sigma}_t^{epi} = \frac{1}{M} \sum_{i=1}^M \left( \hat{q}_i^{obj} \right) \left( \hat{q}_i^{obj} \right)^T - \frac{1}{M^2} \left( \sum_{i=1}^M \hat{q}_i^{obj} \right) \left( \sum_{i=1}^M \hat{q}_i^{obj} \right)^T, \tag{2}$$

where $\hat{q}_i^{obj}$ are the sampled outputs [31]. This formulation also allows us to perform more accurate predictions of the object pose by using the approximate predictive mean

$$\hat{q}_t^{obj} = \frac{1}{M} \sum_{i=1}^M \hat{q}_i^{obj}. \tag{3}$$

**Combining Aleatoric and Epistemic Uncertainty.** We assume independence of the aleatoric and epistemic uncertainties, as in [30, 31]. Therefore, the estimated total uncertainty, using $M$ Monte Carlo dropout samples of $f(\boldsymbol{o}_t)$, is given by $\hat{\boldsymbol{\Sigma}}_t = \hat{\boldsymbol{\Sigma}}_t^{ale} + \hat{\boldsymbol{\Sigma}}_t^{epi}$, resulting in

$$\hat{\boldsymbol{\Sigma}}_t = \frac{1}{M} \sum_{i=1}^{M} \hat{\boldsymbol{\Sigma}}_i^{ale} + \frac{1}{M} \sum_{i=1}^{M} \left(\hat{\boldsymbol{q}}_i^{obj}\right) \left(\hat{\boldsymbol{q}}_i^{obj}\right)^T - \frac{1}{M^2} \left(\sum_{i=1}^{M} \hat{\boldsymbol{q}}_i^{obj}\right) \left(\sum_{i=1}^{M} \hat{\boldsymbol{q}}_i^{obj}\right)^T. \tag{4}$$

### 3.3 Learning the Control Policy

After training the state estimator, we can use it to learn an RL policy $\pi_{est}$ in an environment with visual occlusions. In particular, given an observation of the environment $\boldsymbol{o}_t$, we feed it through our state estimator, using Monte Carlo sampling to approximate the predictive mean object pose $\hat{\boldsymbol{q}}_t^{obj}$ and covariance $\hat{\boldsymbol{\Sigma}}_t$, as shown in Eq. (3) and Eq. (4). Since our state estimator only predicts the object pose, we construct the complete state estimate $\hat{\boldsymbol{s}}_t$ by replacing the object pose in $\boldsymbol{o}_t$ with $\hat{\boldsymbol{q}}_t^{obj}$. Finally, we provide this state estimate $\hat{\boldsymbol{s}}_t$ along with the estimated uncertainty $\hat{\boldsymbol{\Sigma}}_t$ as input to the RL policy, such that $\boldsymbol{a}_t = \pi_{est}(\hat{\boldsymbol{s}}_t, \hat{\boldsymbol{\Sigma}}_t)$. By learning with the state estimator in the loop, $\pi_{est}$ can leverage the uncertainty estimation and better adapt to the inaccuracies of the estimator.

## 4 Experiment Set-up

**Simulated RL Environment.** We develop a planar pushing simulation environment for policy training and data collection using Isaac Sim [34]. In order to speed up the simulation and training, we abstract away the robot as a spherical pusher. We also assume a fixed-size cuboid as the manipulated object. At the start of every episode, we randomly sample the starting object and pusher poses within the workspace. The goal of the episode is to push the object within $1\,\mathrm{cm}$ and $15\,\mathrm{deg}$ of the target position and orientation. We define the reward as $r_t = r_{term} + k_1(1 - d_{trans}) + k_2(1 - d_{rot}) + k_3(1 - v_p)$, where $r_{term}$ is the termination reward (with 50 for reaching the target and $-10$ for violating the workspace boundaries), $d_{trans}$ and $d_{rot}$ are the normalized Euclidean and rotational distances to the target, $v_p$ is the normalized magnitude of the pusher velocity, and $k_1 = 0.1, k_2 = 0.02, k_3 = 0.004$. The control frequency is $15\,\mathrm{Hz}$ with a maximum horizon of $H = 300$ steps or $20\,\mathrm{s}$.

**Sim-to-real Transfer.** To enable zero-shot sim-to-real transfer, we use dynamics randomization on the mass, friction, and restitution. We also add correlated noise, sampled at the beginning of the episode, and uncorrelated noise, sampled at every timestep, to the observation. When considering scenarios with occlusions, they can start at any timestep with a probability of $p = 1/30$. Table 1 summarizes the occlusion duration and observation noise distributions.

| Parameter | Distribution |
|---|---|
| Occlusion Duration | $\mathcal{N}(10, 5^2)$ s |
| Force noise | $\mathcal{N}(0, 0.7^2)$ N |
| Position noise | $\mathcal{N}(0, 0.0025^2)$ m |
| Orientation noise | $\mathcal{N}(0, 0.05^2)$ rad |

Table 1: Occlusion and noise distributions.

**Privileged Policy ($\pi_{priv}$).** We train the privileged policy with Proximal Policy Optimization (PPO) [35] in the occlusion-free environment. The policy takes actions $\boldsymbol{a}_t = [v_x, v_y]$ consisting of the $x$ and $y$ pusher velocities, limited to a maximum $0.05\,\mathrm{m\,s^{-1}}$. We discretize the action space, with 11 bins for each velocity, to use categorical exploration, which improves RL performance for planar pushing [11]. We consider failure if the policy reaches the maximum horizon without completing the task. We use a recurrent architecture for the policy and value networks in PPO to capture the hidden dynamics of the environment.

**State Estimator ($f$).** To train the state estimator, we first deploy 300 uniformly spaced privileged policy checkpoints to collect 750,000 trajectories in the occlusion-free environment. We add observation noise and synthetic occlusions to these trajectories, according to Table 1. For the state estimator, we use a recurrent architecture to capture the temporal dynamics of the environment. The architecture consists of a Long Short-Term Memory (LSTM) layer of size 1024 followed by three linear layers of sizes (512, 256, 8), with tanh non-linearities. We apply dropout with probability 0.2 before every linear layer. In line with [30], we estimate the epistemic uncertainty with 50 stochastic forward passes. The estimator outputs the predicted object pose and aleatoric uncertainty. We

represent the object pose as $\boldsymbol{q}_t^{obj} = (x, y, \sin(\theta), \cos(\theta))$, where $(x, y)$ is the planar position and $\theta$ is the orientation with respect to the $z$ axis. For the uncertainty, the estimator predicts the natural logarithm of the diagonal elements of the covariance matrix $\hat{\boldsymbol{\Sigma}}_t^{ale}$.

**Control Policy ($\boldsymbol{\pi_{est}}$).** We use the same architecture and training procedure for the control policy $\pi_{est}$ as with $\pi_{priv}$, but with occlusions in the training environment. We obtain the state and uncertainty estimates $(\hat{\boldsymbol{s}}_t, \hat{\boldsymbol{\Sigma}}_t)$ through the pre-trained state estimator and input them to $\pi_{est}$.

## 5 Simulation Experiments

### 5.1 Effect of Modeling Uncertainty

We study the effect of modeling aleatoric and epistemic uncertainty on the accuracy of the state estimator. In particular, we compare the performance of our proposed formulation against two alternative formulations: one that removes the Monte Carlo (MC) dropout step, used to model epistemic uncertainty, and another that removes MC dropout and replaces the likelihood loss, used to model aleatoric uncertainty, with a standard Mean Squared Error (MSE) loss. We also include a baseline that uses the visual pose observation without processing. We measure the translational and rotational error of each estimator in a separate test dataset and summarize the results in Table 2.

The results show that modeling aleatoric and epistemic uncertainty significantly improves the state estimator accuracy. The likelihood loss enables the network to predict the aleatoric uncertainty and hence reduce the effect of noisy inputs on the loss, an effect known as learned loss attenuation [30]. MC dropout enables us to sample from the approximate posterior and hence obtain a more accurate pose estimation through the approximate predictive mean. Finally, our state

| Estimator | Trans. Error (mm) | | Rot. Error (deg) | |
|---|---|---|---|---|
| Method | Mean L2 | RMSE | Mean Abs. | RMSE |
| Vision Baseline | 49.83 | 59.65 | 22.56 | 39.61 |
| MSE Loss | 7.62 | 6.49 | 2.51 | 3.37 |
| Likelihood Loss | 6.83 | 5.92 | 2.46 | 3.39 |
| **Ours** | **4.42** | **3.84** | **2.10** | **2.96** |

Table 2: Accuracy of the state estimator without modeling uncertainty (MSE loss), modeling aleatoric uncertainty (likelihood loss), and ours, which models aleatoric and epistemic uncertainty (likelihood loss + MC dropout). The baseline relies only on visual input.

estimator obtains similar or better accuracy than previous model-based approaches [36, 37] in significantly more difficult scenarios. We consider much longer occlusions, with duration $\mathcal{N}(10, 5^2)$ s, higher observation noise, and more complex trajectories involving large object rotations.

### 5.2 Comparison of Policy Training Approaches

We evaluate different techniques to learn the control policy. We consider learning $\pi_{est}(\hat{\boldsymbol{s}}_t, \hat{\boldsymbol{\Sigma}}_t)$, taking as input the state and uncertainty prediction provided by the pre-trained state estimator, as well as $\pi_{est}(\hat{\boldsymbol{s}}_t)$, which only receives the state prediction. We also learn a policy using the variant of our estimator trained with a MSE loss, without capturing uncertainty. Additionally, we consider an end-to-end RL approach, which attempts to learn the policy directly without explicitly learning a state estimator. Furthermore, we evaluate the performance of $\pi_{priv}(\hat{\boldsymbol{s}}_t)$, directly applying the optimal privileged policy with the pre-trained state estimator. Finally, we experiment with a teacher-student behavior cloning approach [27, 28, 29]. We use our privileged policy as the teacher to collect optimal trajectories in the occlusion-free environment, process them to introduce occlusions and noise, and train a student policy to imitate the optimal actions via behavior cloning.

Fig. 2 shows the resulting performance. Our uncertainty-aware policy $\pi_{est}(\hat{\boldsymbol{s}}_t, \hat{\boldsymbol{\Sigma}}_t)$ achieves the highest success rate (94%). Note that, without explicitly receiving the uncertainty as an input, $\pi_{est}(\hat{\boldsymbol{s}}_t)$ achieves slightly lower final performance (92%). We believe this is because, through extensive environment interactions during training, with the state estimator in the loop, it is possible to implicitly obtain information about its uncertainty. Nevertheless, $\pi_{est}(\hat{\boldsymbol{s}}_t)$ still leverages our estimator that captures uncertainty. In fact, when we train the same policy using the MSE estimator, the performance considerably degrades (64%) due to the

lower estimator accuracy. Overall, by modeling aleatoric and epistemic uncertainty we obtain a significantly better estimator and, furthermore, providing the uncertainty estimate explicitly to the policy leads to slightly improved performance.

Applying the optimal privileged policy directly with our state estimator already achieves good success rate (83%). However, including the estimator in the RL training significantly improves performance (94%), indicating the advantage of learning policies that, explicitly or implicitly, account for estimator uncertainty. Finally, end-to-end RL and behavior cloning achieve low success rate, 12% and 38%, highlighting the need to explicitly learn state estimators.

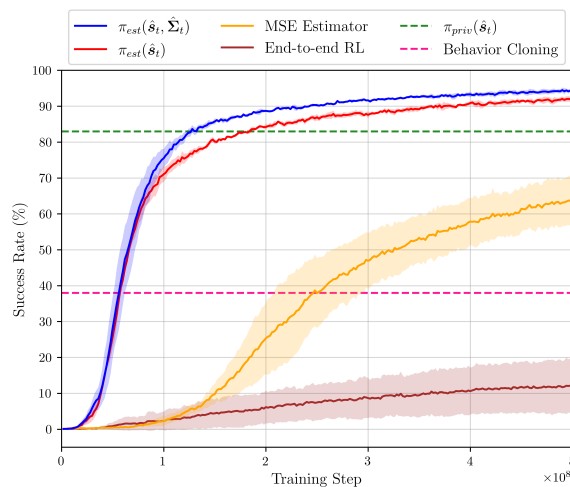

Figure 2: Performance with different training configurations of the control policy. For the RL learning curves, we report mean and standard deviation across three training seeds.

## 5.3  Effect of Occlusion Duration

We evaluate the methods discussed in Section 5.2 with different occlusion durations and show the results in Fig. 3. We report success rate averaged over 2,000 runs with randomized environment configurations. We keep the probability $p = 1/30$ that an occlusion starts at each step but fix the occlusion duration for each scenario and, in the case of full occlusions, lasting until termination. We find that

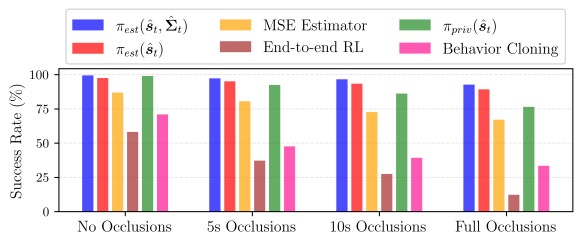

Figure 3: Performance of different control policies with increasing occlusion durations.

$\pi_{est}(\hat{s}_t, \hat{\Sigma}_t)$ and $\pi_{est}(\hat{s}_t)$, the policies trained with our proposed estimator in the loop, achieve minimal performance degradation, 6.7% and 8.3%, between the no and full occlusions scenarios. End-to-end RL and behavior cloning, have the highest decline in performance, 45.8% and 37.6%.

## 5.4  Estimated State and Uncertainty Trajectories

We visualize in Fig. 4 estimated state and uncertainty trajectories generated by $\pi_{est}(\hat{s}_t, \hat{\Sigma}_t)$ and $\pi_{priv}(\hat{s}_t)$ in a full occlusion scenario. For the uncertainty, we use the sum of the predicted variances corresponding to the $(x, y)$ position. $\pi_{est}(\hat{s}_t, \hat{\Sigma}_t)$ manages to solve the task and $\pi_{priv}(\hat{s}_t)$ fails due to violating the workspace boundaries. While $\pi_{priv}(\hat{s}_t)$ immediately makes contact and attempts to rotate the box towards the target, $\pi_{est}(\hat{s}_t, \hat{\Sigma}_t)$ learns to switch contact faces, allowing it to perform a more grad-

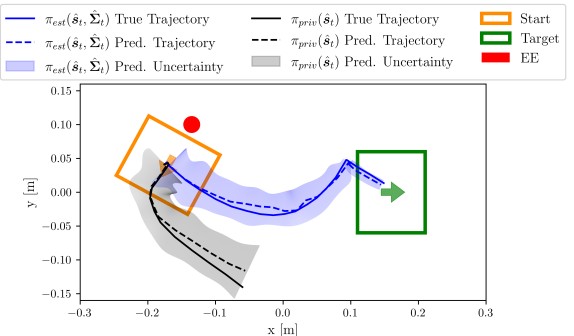

Figure 4: Trajectories generated under full occlusion.

ual rotation, possibly using sticking rather than sliding contact, which creates less uncertainty in the object pose. $\pi_{priv}(\hat{s}_t)$ takes a sharp turn leading to a progressive increase in the estimator uncertainty and error. On the other hand, $\pi_{est}(\hat{s}_t, \hat{\Sigma}_t)$ gradually re-orients the box and again switches contact faces before reaching the target. Notice the progressive decrease in uncertainty during the

initial rotation as the estimator obtains more interaction data, necessary to infer the observation noise and hidden environment dynamics. The final contact-face switch also leads to a dramatic decrease in uncertainty since making contact in a perpendicular face adds significant information about the object pose. Overall, we find that training the control policy with the estimator in the loop leads to different learned behaviors leveraging various strategies to reduce the uncertainty. Further, we quantitatively evaluate the difference in contact switches, an effective uncertainty-reducing strategy, between $\pi_{priv}(s_t)$, the privileged policy in the occlusion-free environment, and $\pi_{est}(\hat{s}_t, \hat{\Sigma}_t)$ in the standard environment with occlusions. We find that $\pi_{priv}(s_t)$ makes $2.68 \pm 1.53$, while $\pi_{est}(\hat{s}_t, \hat{\Sigma}_t)$ makes $4.22 \pm 2.67$ contact switches per episode, averaged across $20,000$ randomized episodes.

## 6 Hardware Experiments

We conduct hardware experiments using a KUKA iiwa robot arm with a F/T sensor at the wrist and a pusher end-effector, as shown in Fig. 1. A camera mounted immediately above the pusher tracks the pose of the manipulated object through AprilTag [18] markers. We use OpTaS [38] to map policy actions (end-effector velocities) to robot joint configurations. Note that this set-up is naturally prone to occlusions as the markers might be outside the camera's field of view or obstructed by the pusher itself. Additionally, we observed significant force readings outside of contact interactions due to the dynamics of the pusher. During training, we neglected these dynamics and instead relied on adding large amounts of correlated and uncorrelated noise to the force observation to provide robustness. For future work, we aim to model and introduce the pusher dynamics in the learning process.

We deploy our proposed policy $\pi_{est}(\hat{s}_t, \hat{\Sigma}_t)$ and state estimator $f(o_t)$ in the robot with zero-shot sim-to-real transfer. We observe similar information-gathering strategies as discussed in Section 5.4. For instance, the policy tends to perform a contact switch, which reduces uncertainty, right before aligning the object with the target, as illustrated in Fig. 4. We evaluate two scenarios with randomized initial configurations: (1) occlusions arising from the onboard perception set-up, and (2) longer human-induced occlusions. In the first scenario, our method achieves $19/20$ successful runs and, in the second, our method achieves $10/10$ with $5\,\mathrm{s}$ occlusions and $7/10$ with full occlusions. The supplementary video shows examples from those scenarios. We find that the state estimator and control policy exhibit good transferability to the physical robot, even in the presence of unseen dynamics.

## 7 Limitations and Conclusion

**Limitations.** Compared to model-based approaches using analytical models, model-free state state estimators require vast amounts of interaction data providing sufficient coverage of the state space. In this work, we use a fixed-shape for the pusher and object, without exploring how to generalize the method to diverse geometries. Additionally, our current simulation set-up neglects any effects of the pusher dynamics on the force measurements, observable during robot experiments. We omit any quantitative evaluation on the accuracy of the estimated uncertainty, which would require a well-characterized system. Furthermore, despite the planar pushing task capturing the key challenges of other non-prehensile manipulation skills, the generalization of our findings to other tasks remains untested. Finally, we only use the estimated uncertainty as an additional input to the control policy; however, we could also apply it to predict failures, trigger fallback recovery mechanisms, and to augment the training dataset for the state estimator, improving the state space coverage.

**Conclusion.** We propose a method to learn visuotactile state estimators for non-prehensile manipulation under occlusions by leveraging privileged policies, learned in simulation, to generate diverse interaction data. Furthermore, we formulate the estimator within a Bayesian deep learning framework, to model its uncertainty, and then train uncertainty-aware RL control policies by incorporating the pre-learned estimator into the RL training loop, both of which lead to significantly improved estimator and policy performance in our extensive simulation experiments. Unlike prior non-prehensile research that relies on complex external perception set-ups to avoid occlusions, our method achieves successful sim-to-real transfer to robotic hardware with a simple onboard camera, under occlusions.

**Acknowledgments**

This work was supported by the JST Moonshot R&D (Grant No. JPMJMS2031), the Kawada Robotics Corporation, and the RAICo Fellows Scheme.

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

# Appendix

## A  Simulation Environment

We develop our planar pushing simulation environment using NVIDIA Omniverse Isaac Sim due to its GPU parallelization capabilities, which significantly accelerate the training and data collection process. Fig. 5 shows a visualization of the simulation environment. The rectangular planar workspace has dimensions $0.6\,\mathrm{m} \times 0.3\,\mathrm{m}$. Note that we designed this workspace based on the maximum reachability in our robotic hardware set-up. For the manipulated object, we use a cuboid of size $0.12\,\mathrm{m} \times 0.1\,\mathrm{m} \times 0.07\,\mathrm{m}$, and for the pusher we use a sphere of radius $0.013\,\mathrm{m}$. We enforce the workspace boundaries with respect to the centroids of the pusher and the object. During policy training and data collection, we randomize the dynamics of the environment, including the mass of the manipulated object as well as friction and restitution coefficients of the table, the pusher, and the object. Table 3 summarizes the randomized dynamics parameters and their corresponding sampling distributions.

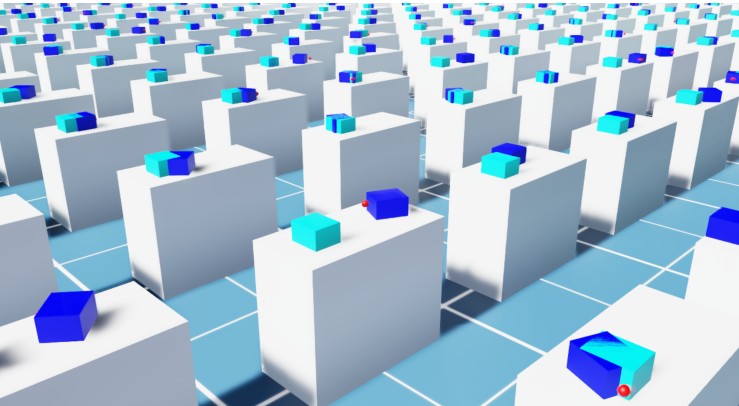

Figure 5: Planar pushing simulation environment in Isaac Sim. The pusher is shown in red, the manipulated object in dark blue, and the target pose in light blue.

| Parameter | Distribution |
|---|---|
| Static friction | $\mathcal{U}(0.3, 0.5)$ |
| Dynamic friction | $\mathcal{U}(0.1, 0.3)$ |
| Restitution | $\mathcal{U}(0.1, 0.7)$ |
| Object mass | $\mathcal{U}(3.0, 3.5)$ kg |

Table 3: Dynamics randomization parameters and corresponding sampling distributions.

## B  Additional Training Details

### B.1  Reinforcement Learning Policies

We process the RL observation by scaling each component to the range $[-1, 1]$. In particular, we scale the $(x, y)$ coordinates using the workspace dimensions and for the object orientation $\theta$ we use $\sin(\theta)$ and $\cos(\theta)$. For the pusher force $\boldsymbol{f}_t^e$, we apply $\mathrm{clip}(\boldsymbol{f}_t^e, -10, 10)/10$ component-wise. In practice, this means that we limit the magnitude of the force reading along each axis to $10\,\mathrm{N}$. When providing the predicted uncertainty from the state estimator as an RL policy observation, we use the standard deviations, clipped to the range $[0, 1]$. Note that in these cases where the RL policy receives the predicted uncertainty, we keep the reward function unchanged.

During RL policy training, we use a learning rate scheduler based on the KL divergence of the policy, as in [39, 22]. The scheduler has a target KL divergence of $7 \cdot 10^{-3}$, a minimum learning rate of $1.5 \cdot 10^{-4}$ and a maximum learning rate of $10^{-2}$. For the policy function, we use a neural network architecture with the following layers and corresponding sizes: linear (128) + LSTM (256) + linear (128) + linear (22), with $\texttt{tanh}$ nonlinearities. The output consists of 22 logits that define the categorical distributions over the $x$ and $y$ velocities. For the value function, we use the same neural network architecture but replace the final linear (22) layer with a linear (1) layer that outputs the state value prediction. During training, when episodes reach the maximum horizon, thereby terminating, we use the value function prediction corresponding to the final observation to bootstrap the final reward. Table 4 summarizes the remaining RL hyperparameters for PPO [35] training. Finally, Fig. 6 shows the training performance of the privileged policy $\pi_{priv}(\boldsymbol{s}_t)$ in the occlusion-free environment.

| Hyperparameter | Value |
|---|---|
| Rollout Steps | 100 |
| Parallel Environments | 4000 |
| Mini-batch Size | 25000 |
| Epochs | 5 |
| Clip Range ($\epsilon$) | 0.2 |
| Discount Factor ($\gamma$) | 0.993 |
| GAE Parameter ($\lambda$) | 0.95 |
| Entropy Loss Coefficient | 0.01 |
| Value Loss Coefficient | 1.0 |

Table 4: PPO hyperparameters.

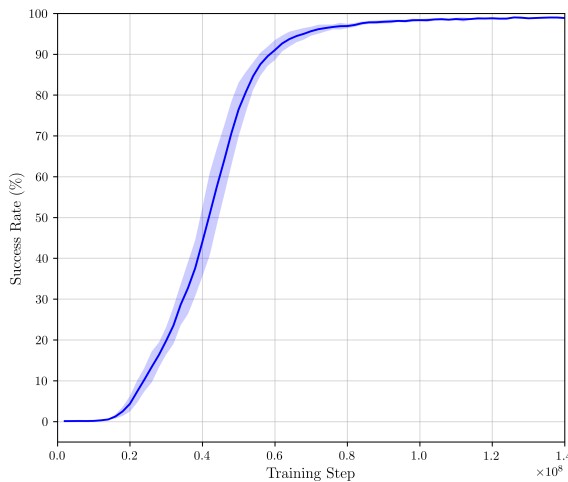

Figure 6: Training performance of the privileged policy $\pi_{priv}(\boldsymbol{s}_t)$. We report mean and standard deviation across three random seeds.

## B.2 State Estimator

To train and evaluate the state estimators, we collect separate training, validation, and testing datasets containing $7.5 \cdot 10^5$, $1.5 \cdot 10^5$ and $1.5 \cdot 10^5$ trajectories respectively. We process the trajectories scaling the pose and force measurements to the range $[-1, 1]$ as discussed for the RL training in Appendix B.1. Additionally, Table 5 summarizes the training hyperparameters for the state estimator.

Fig. 7 and Fig. 8 show the validation loss when training the state estimator with the likelihood loss and the MSE loss, respectively.

| Hyperparameter | Value |
|---|---|
| Mini-batch Size | 30000 |
| Epochs | 110 |
| Learning Rate | $10^{-3}$ |
| Optimizer | Adam [40] |
| Sequence Length | 300 |

Table 5: State estimator hyperparameters.

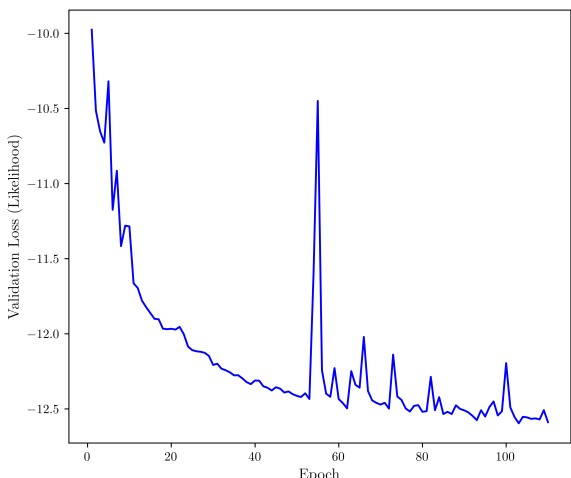

Figure 7: Estimator validation loss when training with the likelihood loss.

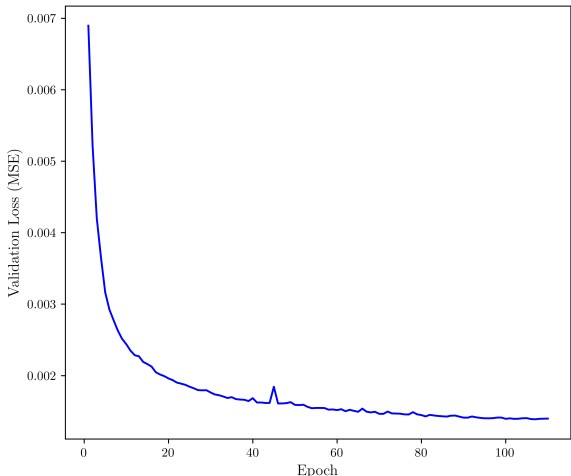

Figure 8: Estimator validation loss when training with the MSE loss.

### B.3 Behavior Cloning Baseline

To train the behavior cloning baseline discussed in Section 5.2, we collect new training and validation datasets using the last privileged policy checkpoint to provide optimal trajectories. We do not use a testing dataset since we evaluate the baseline directly in the planar pushing simulation environment. The training and validation datasets contain $7.5 \cdot 10^5$ and $1.5 \cdot 10^5$ trajectories respectively. We

process the trajectories to scale the pose and force measurements, add sensory noise, and introduce occlusions using the same procedure as for the state estimator.

The neural network architecture for the behavior cloning policy is the same as for the RL policy function, discussed in Appendix B.1. Hence, the output in both cases consists of 22 logits that define separate categorical distributions over the $x$ and $y$ velocities. We train the behavior cloning policy using a loss function defined as the sum of the cross-entropy between the predicted and target distributions for the $x$ and $y$ velocities. The training hyperparameters are the same as shown in Table 5. Finally, Fig. 9 shows the validation loss during training.

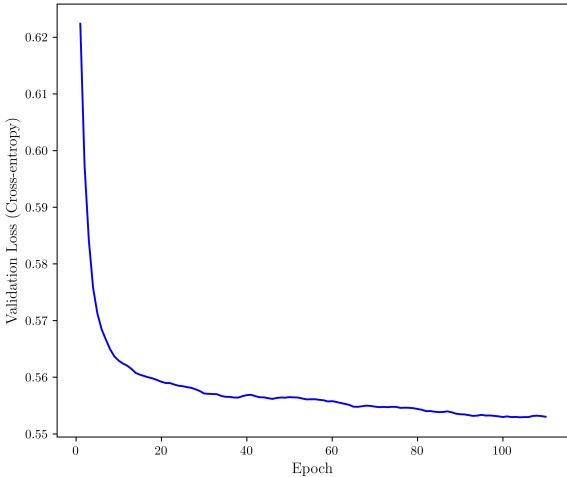

Figure 9: Behavior cloning policy validation loss.

