# OpenReview forum: "Learning Visuotactile Estimation and Control for Non-prehensile Manipulation under Occlusions"
_robot-learning.org/CoRL/2024/Conference — CoRL 2024_

### Official Review · Reviewer_eXyy · 2024-07-13
**Contribution to solving non-prehensile manipulation learning**

**Originality:** 3
**Technical Quality:** 4
**Clarity Of Presentation:** 4
**Potential Impact:** 3
**Recommendation:** 3
**Confidence:** 4

**Review:**

The paper tackles a relevant problem in its aim to learn manipulation that is robust against adverse events such as occlusions. It is well written and comprehensible. The experimental evaluation is nice as it tests the approach's robustness and suitable shows the advantages offered by Bayesian modeling in behavior generation.

I just have one point of critique, but it is an important one. The paper could to better in contrasing itself against the existing related work. The related work under section "Non-prehensile Manipulation"  the authors refer to substantial works but do not exactly make by which criteria they contrast against them.

The authors state that they contrast against work that uses only external (eye-on-hand) setups, and I can see this distinction. I do not know of work that also uses an eye-in-hand setup as the authors. However, this criterion is convolved with that of "tracking". The authors also use a tracking framework as other works (using markers). As it is not a criterion for contrasing against the related work it is not useful in this paragraph.

There are approaches with external cameras that can cope with occlusions too (see below). It is unclear if an eye-in-hand setup is actually a sufficient criterion to constrast substantially against these other works. One could argue that external camera setups suffer even more substantially from occlusions due to the robot arm, or at least from different kinds of occlusions.

Related work that the authors might consider adding to their discussion:

Lee, Michelle A., Yuke Zhu, Krishnan Srinivasan, Parth Shah, Silvio Savarese, Li Fei-Fei, Animesh Garg, and Jeannette Bohg. "Making sense of vision and touch: Self-supervised learning of multimodal representations for contact-rich tasks." In 2019 International conference on robotics and automation (ICRA), pp. 8943-8950. IEEE, 2019.

Lee, Michelle A., Brent Yi, Roberto Martín-Martín, Silvio Savarese, and Jeannette Bohg. "Multimodal sensor fusion with differentiable filters." In 2020 IEEE/RSJ International Conference on Intelligent Robots and Systems (IROS), pp. 10444-10451. IEEE, 2020.

Yu, Kuan–Ting, and Alberto Rodriguez. "Realtime state estimation with tactile and visual sensing. application to planar manipulation." In 2018 IEEE International Conference on Robotics and Automation (ICRA), pp. 7778-7785. IEEE, 2018.

**Quality Of The Limitations Section:**

3

**Questions For Rebuttal:**

- The related work section on non-prehensile manipulation should be revised to more clearly contrast against related work and potentially signify why the switch to eye-in-hand is relevant.
-- Rephrase: L106 -- collecting real world data is not prohibitive, others have done it! See, e.g.:

Bauza, Maria, Ferran Alet, Yen-Chen Lin, Tomás Lozano-Pérez, Leslie P. Kaelbling, Phillip Isola, and Alberto Rodriguez. "Omnipush: accurate, diverse, real-world dataset of pushing dynamics with rgb-d video." In 2019 IEEE/RSJ International Conference on Intelligent Robots and Systems (IROS), pp. 4265-4272. IEEE, 2019.

**Robotics Focus:**

4

**Summary Of Paper:**

The authors propose a method for learning prehensile manipulation that positions cubes in a table-top scenario. The method uses a Bayesian state estimator to assess uncertainty in the state and learns a policy that makes use of these state estimates including uncertainty. In comparison to non-probabilistic baselines the method performs superior and can cope with occlusions to the visual camera input.

**Summary Of Recommendation:**

Good paper. The paper makes a useful contribution and is sufficiently different from the related work, but should include further works and improve the way it contrasts against the related work.

---

### Official Review · Reviewer_XGFC · 2024-07-19
**The method addresses the challenge of occlusions in non-prehensile manipulation by using visuotactile fusion.**

**Originality:** 4
**Technical Quality:** 3
**Clarity Of Presentation:** 4
**Potential Impact:** 3
**Recommendation:** 3
**Confidence:** 4

**Review:**

Strengths:

a. ​Addressing Occlusions:
The method effectively addresses the challenge of occlusions in non-prehensile manipulation by using visuotactile fusion.

b. Bayesian Deep Learning Framework:
Using a Bayesian framework to model uncertainty of the real world enhances the robustness of the control.

c. Simulation-to-Real-World:
The paper presents good real-world experiments and results.

d. Diverse Dataset Generation:
By generating diverse interaction data through privileged policies.


Weaknesses:

a. ​Fixed Pusher and Object Shapes:
The study uses fixed shapes for the pusher and objects, limiting the generalizability of the method to various geometries and sizes in different applications.

b. Ignoring Pusher Dynamics:
The current simulation setup ignores the dynamics of the pusher on force measurements.

c. Lack of Quantitative Uncertainty Evaluation:
The study lacks a detailed quantitative evaluation of the estimated uncertainty.

d. Insufficient Generalization Testing:
The study has not been tested on other complex tasks, limiting its broader application.

**Quality Of The Limitations Section:**

2

**Questions For Rebuttal:**

1. The paper mentions using a Bayesian deep learning framework to capture uncertainty, which significantly improves the accuracy of the estimation. Could you explain the specific reasons for choosing the Bayesian framework? What are the distinct advantages of this method in paper's application compared to other possible choices, such as traditional neural networks or alternative uncertainty modeling methods?

2. In the experiments, the paper shows performance differences between different control strategies, particularly those that incorporate the pre-trained state estimator in the RL training. Could you further explain the reasons for the performance differences, especially why the paper's strategy (e.g., $π_{est}(\hat{s}_t, \hat{Σ̂}_t)$) performs best in handling visual occlusions?

3. The paper claims successful sim-to-real transfer, particularly under conditions of occlusions and sensor noise. Could you provide more detailed descriptions of the specific strategies employed to ensure this successful transfer? How does the method effectively maintain its performance when dealing with unknown dynamics and noise in real-world robotic hardware setups?

**Robotics Focus:**

4

**Summary Of Paper:**

The main idea of this paper is to develop a method that uses visuotactile fusion for state estimation and control to address the challenge of occlusions in non-prehensile manipulation tasks. The proposed approach leverages privileged policies trained in a simulation environment to generate diverse interaction data. This data is then used within a Bayesian deep learning framework to model uncertainty in state estimation. The pre-trained state estimator is integrated into the reinforcement learning (RL) training loop to develop control policies that can handle uncertainty effectively.

**Summary Of Recommendation:**

The method effectively addresses the challenge of occlusions in non-prehensile manipulation by using visuotactile fusion.

---

### Official Review · Reviewer_TfpZ · 2024-07-20

**Originality:** 2
**Technical Quality:** 3
**Clarity Of Presentation:** 3
**Potential Impact:** 3
**Recommendation:** 3
**Confidence:** 4

**Review:**

Strengths:
- The paper is well-written, tackles the important problem of control for occlusion-robust visuotactile manipulation, and demonstrates the method on hardware.

Concerns:
- One key assumption that the paper makes is that the state distribution covered by executing the privileged policy covers the subset of the state space that needs to be visited in the presence of occlusions, as otherwise, the state estimator will be OOD in those states. However, this is not always the case - in many POMDPs, information gathering actions must be taken to reduce state uncertainty by visiting non-occluded states. Such states don’t need to be visited by the privileged policy. It would help to visualize the state distribution of the privileged policy, and compare with the state distribution of the uncertainty-aware policy, and to discuss why the privileged data provides sufficient coverage for the pusher example. Also, how should the method be adapted to the case where the privileged data is insufficient, and how can one detect this?
- Line 118: based off the experiments, where an RNN is used, the state estimator actually uses a history of environment observations instead of just the current observation, correct? It would help to make this explicit in the methods section, as this makes a difference: if the state estimator is only allowed to use the instantaneous observation, which is fully occluded, there is no way for the estimator to distinguish between two states that lead to identical observations due to occlusions, whereas giving the estimator a history allows some ability to distinguish based off of the history of actions and where the object was prior to occlusion.
- Some qualitative discussion on what kind of information-gathering is done by the uncertainty-aware policy would be helpful in convincing the reader of the method’s utility in solving visuotactile POMDPs more generally.
- The paper does not compare to any existing methods for belief-space planning under occlusions [1], extensions of teacher-student RL to the partially observed case [2], or uncertainty-aware visual control more generally [3]. Baseline comparisons to a subset of these methods would help to better place the method in the literature.

[1] Patil et al. Gaussian Belief Space Planning with Discontinuities in Sensing Domains. ICRA 2014.
[2] Shenfeld et al. TGRL: An Algorithm for Teacher Guided Reinforcement Learning. ICML 2023.
[3] Chou et al. Safe Output Feedback Motion Planning from Images via Learned Perception Modules and Contraction Theory. WAFR 2022.

**Quality Of The Limitations Section:**

2

**Questions For Rebuttal:**

Please see review

**Robotics Focus:**

4

**Summary Of Paper:**

- The paper proposes a method for occlusion-robust visuotactile manipulation. The algorithm first trains a fully-observed policy with privileged information, renders imperfect/partial observations from this data, and uses this rendered data to train a state estimator, done via Bayesian deep learning. This state estimator is given as input to train a partially-observed RL policy. The method is evaluated on a planar pusher example with occlusions.

**Summary Of Recommendation:**

Overall, I recommend a weak accept: the paper is generally solid, but some more discussion on the limitations and comparison to existing uncertainty-aware visual control methods would strengthen the paper.

---

### Author Rebuttal · Authors · 2024-08-13

We thank all of the reviewers for the insightful comments. We have uploaded a zip file with the revised version of the manuscript as well as a version (diff) highlighting the changes, with a small red font corresponding to the deleted text and larger blue font corresponding to the added text.

---

### Decision · Program_Chairs · 2024-09-04

**Decision:**

Accept

**Comment:**

Update: The reviewers have addressed important concerns that reviewers have raised. In particular, there was an important point about how privileged policy would have a different state distribution than the non-privileged policy. From what I have understood, the authors address this by keeping the states that were encountered by the intermediate checkpoints when training the privileged policy. I do not think this is the fundamental solution to the state OOD problem, it is an okay answer. I also believe that there several interesting aspects to the paper, such as Bayesian formalization and learning of state estimator, which was not covered in previous approaches.

======================

This paper presents a method for non-prehensile manipulation that is robust to occlusion, via the use of visuotactile sensing.

Overall, reviewers are positive about the hardware experiment, the problem it tackles, and the method it proposes. Some technical points need to be ironed out (i.e. how the privileged policy isn't encouraged to take information-gathering actions, justifications on certain technical decisions, such as why the use of Bayesian framework, etc.).

Several closely related works seem to have gone undiscussed. In addition to the ones reviewer eXyy suggested, I suggest readers to check the following papers:
- CORN: Contact-based Object Representation for Nonprehensile Manipulation of General Unseen Objects
Cho et al., ICLR 2024
- Pre- and Post-Contact Policy Decomposition for Non-Prehensile Manipulation with Sim-to-Real Transfer.
Kim et al., IROS 2023